# The Assessment of the Efficacy of Imperatorin in Reducing Overactive Bladder Symptoms

**DOI:** 10.3390/ijms242115793

**Published:** 2023-10-31

**Authors:** Paulina Iwaniak, Piotr Dobrowolski, Jan Wróbel, Tomasz Kluz, Artur Wdowiak, Iwona Bojar, Klaudia Stangel-Wójcikiewicz, Ewa Poleszak, Artur Jakimiuk, Marcin Misiek, Łukasz Zapała, Andrzej Wróbel

**Affiliations:** 1Department of Experimental and Clinical Pharmacology, Medical University of Lublin, Jaczewskiego 8b, 20-090 Lublin, Poland; 2Department of Functional Anatomy and Cytobiology, Faculty of Biology and Biotechnology, Maria Curie-Sklodowska University, Akademicka 19, 20-033 Lublin, Poland; 3Medical Faculty, Medical University of Lublin, 20-093 Lublin, Poland; wrobeljan@onet.eu; 4Department of Gynecology, Gynecology Oncology and Obstetrics, Institute of Medical Sciences, Medical College of Rzeszow University, Rejtana 16c, 35-959 Rzeszow, Poland; jtkluz@interia.pl; 5Department of Obstetrics and Gynecology, Faculty of Health Sciences, Medical University of Lublin, Staszica 4-6, 20-081 Lublin, Poland; wdowiakartur@gmail.com; 6Department of Women’s Health, Institute of Rural Health in Lublin, ul. Jaczewskiego 2, 20-090 Lublin, Poland; iwonabojar75@gmail.com; 7Department of Gynecology and Oncology, Jagiellonian University Medical College, M. Kopernika 23, 31-501 Kraków, Poland; ksw@cm-uj.krakow.pl; 8Department of Applied and Social Pharmacy, Laboratory of Preclinical Testing, Medical University of Lublin, Chodźki 1, 20-093 Lublin, Poland; ewapoleszak@umlub.pl; 9Department of Obstetrics and Gynecology, National Medical Institute of the Ministry of Interior and Administration, Wołoska 137, 02-507 Warsaw, Poland; jakimiuk@yahoo.com; 10Center for Reproductive Health, Institute of Mother and Child, Kasprzaka 17a, 01-211 Warsaw, Poland; 11Department of Gynecologic Oncology, Holy Cross Cancer Center, 25-377 Kielce, Poland; marcin.misiek@onkol.kielce.pl; 12Clinic of General, Oncological and Functional Urology, Medical University of Warsaw, Lindleya 4, 02-005 Warsaw, Poland; lzapala@wum.edu.pl; 13Second Department of Gynecology, Medical University of Lublin, Jaczewskiego 8, 20-090 Lublin, Poland; wrobelandrzej@yahoo.com

**Keywords:** overactive bladder, imperatorin, rats cystometry, retinyl acetate

## Abstract

Overactive bladder syndrome (OAB) is a prevalent condition that affects the elderly population in particular and significantly impairs quality of life. Imperatorin, a naturally occurring furocoumarin, possesses diverse pharmacological properties that warrant consideration for drug development. The aim of this study was to investigate the potential of imperatorin (IMP) to attenuate the cystometric and biochemical changes typically associated with retinyl acetate-induced overactive bladder (OAB) and to assess its viability as a pharmacological intervention for OAB patients. A total of 60 rats were divided into four groups: I—control, II—rats with rapamycin (RA)-induced OAB, III—rats administered IMP at a dose of 10 mg/kg/day, and IV—rats with RA-induced OAB treated with IMP. IMP or vehicle were injected intraperitoneally for 14 days. The cystometry and assessment of bladder blood flow were performed two days after the last dose of IMP. The rats were then placed in metabolic cages for 24 h. Urothelial thickness measurements and biochemical analyses were performed. Intravesical infusion of RA induced OAB. Notably, intraperitoneal administration of imperatorin had no discernible effect on urinary bladder function and micturition cycles in normal rats. IMP attenuated the severity of RA-induced OAB. RA induced increases in urothelial ATP, calcitonin gene-related peptide (CGRP), organic cation transporter 3 (OCT3), and vesicular acetylcholine transporter (VAChT), as well as significant c-Fos expression in all micturition areas analyzed, which were attenuated by IMP. Furthermore, elevated levels of Rho kinase (ROCK1) and VAChT were observed in the detrusor, which were reversed by IMP in the context of RA-induced OAB in the urothelium, detrusor muscle, and urine. Imperatorin has a mitigating effect on detrusor overactivity. The mechanisms of action of IMP in the bladder appear to be diverse and complex. These findings suggest that IMP may provide protection against RA-induced OAB and could potentially develop into an innovative therapeutic strategy for the treatment of OAB.

## 1. Introduction

Overactive bladder (OAB) is a disorder of the lower urinary tract characterized primarily by a strong urge to urinate and sometimes accompanied by incontinence, frequent urination, and nocturia. Current pharmacotherapy options for OAB are considered inadequate, leading researchers to explore new treatment options, especially for patients who do not fully respond to standard therapies such as antimuscarinics, β3-agonists, and intravesical botulinum toxin A injection [1,2,3,4].

There is growing interest among researchers in the potential medical applications of natural products, including plants and plant-derived substances, due to their antioxidant, anticancer, and antibacterial properties [5,6].

Imperatorin, a natural furocoumarin found in plants of the *Apiaceae* and *Rutaceae* families, is widely used in Traditional Chinese Medicine (TCM) for various purposes, including swelling reduction, pain relief, wound healing, diabetes treatment, and gastrointestinal disorders [7,8,9,10,11,12,13]. It exhibits promising pharmacological properties suitable for drug development, possessing antioxidant, immunomodulatory, anticancer, and neuroprotective properties, making it a potential candidate for the treatment of gynecological, cardiovascular, cerebrovascular, neurological, inflammatory, microbial, and cardiopulmonary diseases [5,14,15,16,17,18]. The primary pharmacological activities of imperatorin appear to be anti-inflammatory and neuroprotective. Imperatorin was evaluated for its potential inhibition of macrophage functions involved in inflammation, including suppression of cyclooxygenase-1 (COX-1), cyclooxygenase-2 (COX-2), and lipoxygenase (5-LOX) enzymes, and reduction of prostaglandin 2 (PGE2), leukotriene 4 (LTC4), and nitric oxide (NO) synthesis induced by *E. coli* lipopolysaccharides (LPS). Imperatorin showed significant inhibitory effects on 5-LOX release (IC50 < 15 µmol), PGE2 release (25–100 µmol), LTC4 generation (1–100 µmol), and LPS-induced PGE2 release, along with NO synthesis (IC50 = 9.2 µmol), possibly related to COX-2 activity. Furanocoumarins, such as imperatorin, hold potential for the prevention and treatment of central nervous system (CNS) disorders affecting learning, anxiety, and epilepsy. Imperatorin has been shown to increase gamma-aminobutyric acid (GABA) levels, which play a critical role in CNS sedation, anxiety reduction, and antiepileptic effects. In addition, it has been studied as a potential acetylcholinesterase (AChE) inhibitor, acting on acetylcholine receptors (AChR) involved in cognitive functions. This inhibition may have implications for neurodegenerative diseases such as Alzheimer’s and Parkinson’s, suggesting a neuroprotective role for imperatorin [19,20,21,22]. Furthermore, Mendel et al. demonstrated the potent myorelaxant activity of imperatorin in isolated rat jejunal strips. Imperatorin induced a progressive myorelaxation in the dose range of 0.001–100 µmol, with relaxation comparable to isoproterenol at 0.1 µmol. The mechanism of action may involve several Ca^2+^ influx pathways. Imperatorin exhibits diverse pharmacological properties with potential applications in traditional medicine and drug development, particularly in the areas of inflammation, neuroprotection, and myorelaxation. Previous studies have already shown promising effects of various herbal medicines in the treatment of overactive bladder, including gosha-jinki-gan, hachimi-jio-gan, buchu, cornsilk, cleavers, and horsetail [23,24]. However, the clinical effectiveness of these herbal remedies remains inconclusive and requires further clinical testing. Nevertheless, herbal medicines are generally well tolerated by patients due to their low incidence of side effects [2,25]. Dent et al. (2020) assert that imperatorin regulates the expression of numerous genes and proteins, such as interleukin-6 (IL-6), B-cell lymphoma 2 (Bcl-2), apoptosis regulator (BAX), induced myeloid leukemia cell differentiation protein (Mcl-1), caspase-3, AChE, PGE2, and peroxisome proliferator-activated receptor gamma (PPARγ). Imperatorin also impacts various signaling pathways, including PI3K/Akt, MAPK, NF-κB, Nrf2/HO-1, IKK/IjB/NF-jB, and ERK. Owing to these effects, imperatorin can be utilized in the treatment of various diseases [26]. Therefore, we undertook an evaluation of the effects of imperatorin on urinary bladder function, both in the normal state and in a model of detrusor overactivity. Imperatorin appears to be an interesting herbal medicine that could potentially alleviate lower urinary tract symptoms associated with overactive bladder.

The aim of this study was to investigate whether imperatorin could counteract the cystometric and biochemical changes induced by rapamycin (RA) that are typically associated with overactive bladder. In doing so, the study aimed to determine whether this natural compound could serve as a viable pharmacological approach for the treatment of OAB patients.

## 2. Results

### 2.1. Characteristics of Cardiovascular and Diuresis Parameters in Experimental Groups

The thickness of urothelium was comparable in all examined groups. There was no significant effect of RA and IMP on bladder blood flow (BBF) and urine production (UP). In addition, assessment of cardiovascular function showed no significant effect of RA and IMP on arterial pressure (MAP) and heart rate (HR) (Figure 1).

### 2.2. The Influence of Imperatorin on Retinyl Acetate-Induced Changes in Cystometric Parameters

Based on the conscious cystometry parameters, it was found that RA injection significantly increased ANVC (amplitude of nonvoiding contractions), AUC (area under the curve), DOI (duration of intercontraction intervals), FNVC (frequency of nonvoiding contractions), and BP (bladder pressure) compared to the control group. Conversely, it significantly decreased the values of VV (voided volume), VTNVC (voiding efficiency), ICI (intercontraction interval), TP (threshold pressure), and BC (bladder capacity) parameters, while imperatorin restored these parameters to levels seen in the control group. However, the therapeutic effect of imperatorin on the ICI parameter was less pronounced. Interestingly, in rats who received imperatorin together with retinyl acetate, a major drop in BP, DOI, ANVC, FNVC, and AUC was observed when compared to the RA group. Moreover, administration of imperatorin to RA rats resulted in a significant increase in TP, VV, ICI, BC, and VTNC. However, administration of imperatorin effectively counteracted the effects of RA in all the mentioned parameters, although the reversal was not complete for DOI and FNVC. No statistical differences in PVR (post-void residual) and MVP (maximum voiding pressure) parameters were observed (Figure 2).

### 2.3. Selected Biochemical Analysis

Interestingly, similar observations were made in the biochemical analysis performed. RA exerted a potent effect by significantly increasing the levels of VAChT (vesicular acetylcholine transporter), TRPV1 (transient receptor potential cation channel subfamily V member 1), ROCK (Rho-associated protein kinase), OCT3 (organic cation transporter 3), NIT (nitric oxide), NGF (nerve growth factor), MAL (myelin-associated glycoprotein), CGRP (calcitonin gene-related peptide), BDNF (brain-derived neurotrophic factor), and ATP (adenosine triphosphate) compared to controls (Figure 3). However, when RA was co-administered with IMP, a complete reversal of the effects of RA was observed in TRPV1, ROCK, NIT, and CGRP parameters, and the remaining biochemical parameters were almost two-fold decreased by imperatorin administration (Figure 3).

### 2.4. Analysis of c-Fos—A Neuronal Activity

Analysis of c-Fos expression corroborated the findings observed in the biochemical parameters (Figure 4). RA administration significantly increased the levels of c-Fos in MPA (medial preoptic area), PMC (pontine micturition center) and vIPAG (ventrolateral periaqueductal gray) compared to all other groups. However, co-administration of imperatorin effectively reversed this effect in the case of PMC and vIPAG and showed a slightly smaller but still curative effect in the case of MPA (Figure 4).

## 3. Discussion

Advances in diagnosis and innovative approaches to the treatment of urinary disorders have greatly expanded the options available to improve the care of patients with overactive bladder (OAB) compared to just a few years ago. These developments are aimed at improving patients’ quality of life and relieving their symptoms. There has been a growing focus on natural products, which have demonstrated multiple pharmaceutical effects and have become a vibrant field for novel drug discovery due to their ability to interact with multiple biological targets. Over the past three decades, the consumption of herbal medicinal products and supplements has increased significantly, with at least 80% of individuals worldwide incorporating them into their primary care. While therapies utilizing these agents hold promising potential and the effectiveness of various herbal products has been established, several products remain unproven, and their use is either poorly monitored or not monitored at all. Adverse events arising from the intake of herbal medicines can be attributed to various causes, such as the incorrect usage of incorrect botanical species, contamination of herbal products with undeclared drugs, hazardous or toxic substances, overconsumption, misuse of herbal medicines by healthcare providers or users, and concomitant use of herbal medicines with other medications [27].

Previous studies have highlighted the potential of imperatorin, a naturally occurring furanocoumarin. Imperatorin has shown a wide range of therapeutic properties, including anti-inflammatory, analgesic, anticancer, neuroprotective, antibacterial, antiosteoporotic, antioxidant, myorelaxant, cardioprotective, hepatoprotective, and antiviral effects. These diverse properties suggest that imperatorin could be a valuable treatment option for gynecological, cardiovascular, cerebrovascular, and neurological diseases [5,28,29,30,31,32,33,34].

It is noteworthy that our study did not identify any adverse effects of imperatorin at the dose (10 mg/kg for 14 days in succession) administered, which was four times less than the dose referenced in prior literature [5,26]. We deliberately opted for this lower dose to ensure safety and decrease the likelihood of potential side effects in our experimental configuration. Although the lack of side effects is a noteworthy discovery, it is crucial to recognize that our findings correspond with the objective of discovering secure treatment choices.

Until now, there has been a lack of data on the influence of imperatorin on urinary bladder function in patients with overactive bladder (OAB). Therefore, the aim of our study was to evaluate the effect of imperatorin on urinary bladder activity using an animal model of OAB induced by retinyl acetate. Our experiment was complex because the urothelium and the detrusor muscle play a crucial role in the regulation of bladder function. In addition, there are several neurotransmitter-related changes in overactive bladder.

In our current study, our primary focus was to investigate the effects of imperatorin on cystometric and biochemical parameters related to bladder function, both in the normal state and in a model simulating detrusor overactivity. We induced detrusor overactivity by transient intravesical infusion of 0.75% retinyl acetate, a common feature observed in patients with overactive bladder. It is worth noting that retinyl acetate did not induce significant histopathological inflammatory lesions in the bladder wall, unlike models induced by cyclophosphamide or acetic acid, making it a valuable choice for our OAB induction model [35].

The most important finding of our study is that 14 days of intraperitoneal (i.p.) administration of imperatorin at a dose of 10 mg/kg/day effectively normalized the pathological changes in involuntary bladder behavior and reduced the severity of detrusor overactivity induced by intravesical retinyl acetate instillation. Importantly, imperatorin had no discernible effect on bladder function under healthy conditions (see Figure 1). These results suggest that i.p. treatment with imperatorin may serve as a potential adjunctive therapy to standard treatments for overactive bladder.

In the context of retinyl acetate (RA)-induced detrusor overactivity, administration of imperatorin resulted in significant changes in several key parameters, as shown in Figure 2. These changes included an increase in ICI, TP, BC, VV, and VTNVC. There was also a decrease in ANVC, FNVC, AUC, BP, and DOI.

These changes in cystometric parameters suggest that imperatorin effectively suppresses basal detrusor muscle tone and excitability during the storage phase of the micturition cycle. Importantly, the results of our study indicate that imperatorin at a dose of 10 mg/kg/day does not adversely affect MVP or PVR. This dual effect demonstrates that imperatorin not only improves the storage phase but also maintains voiding efficiency. A direct confirmation of imperatorin’s effect on detrusor stability is its ability to reduce DOI, a common indicator in in vivo studies used to diagnose overactive bladder, as unstable detrusor contractions typically occur during the storage phase in OAB [36].

Imperatorin has demonstrated the ability to increase VTNVC, as shown in Figure 2. In rodent cystometry, VTNVC serves as an analogous measure to the volume at the onset of the first involuntary detrusor contraction in humans. This parameter, VTNVC, is highly reliable for assessing the efficacy of treatments for overactive bladder (OAB). Its importance lies in its correlation with a reduction in the frequency of urinary incontinence episodes and a reduction in the frequency of micturition [37].

In earlier studies, imperatorin was investigated for its potential as an antihypertensive and cardioprotective agent. It was found that imperatorin at a dose of 15 mg/kg stimulated endothelial nitric oxide synthase, an enzyme responsible for producing nitric oxide, which plays an important role in vasodilation. This mechanism led to a reduction in blood pressure and suppression of cardiac hypertrophy, as well as prevention of degenerative changes in myocardial tissue [38,39].

However, in our current study, we administered imperatorin at a dose of 10 mg/kg/day and observed that it had no significant effect on urine production, heart rate, or blood pressure (as shown in Figure 1). This lack of effect on these parameters can be considered a positive result, particularly for patients with overactive bladder. Excessive urine output can exacerbate lower urinary tract symptoms and increase detrusor overactivity. Conversely, our findings may have important implications for patients with overactive bladder and other comorbidities, such as cardiovascular disease, as they suggest that imperatorin can be used safely without adverse effects on urinary function or blood pressure.

Disturbances in bladder wall perfusion can potentially lead to bladder dysfunction, mainly due to structural changes in the urothelium, detrusor muscle, and ischaemic nerve damage [40]. However, in our experimental observations, we did not observe any significant changes in urothelium thickness or bladder blood flow after administration of imperatorin and retinyl acetate (as shown in Figure 1).

This suggests that the effects of imperatorin on bladder function, as investigated in our study, may not be associated with changes in urothelium thickness or bladder blood flow, highlighting the need for further investigation into the specific mechanisms by which imperatorin affects bladder function.

Previous research conducted by Mendel et al. in 2015 investigated the remarkable myorelaxant properties of imperatorin, specifically its role as a calcium antagonist acting on vascular smooth muscle in isolated rat jejunum strips. The study showed that in the dose range of 0.001–100 µmol, imperatorin exhibited a gradual myorelaxing effect. Remarkably, at concentrations of 25 and 50 µmol, imperatorin induced a relaxation comparable in magnitude to the response induced by isoproterenol at a concentration of 0.1 µmol. Isoproterenol is a drug known for its ability to relax smooth muscle and is structurally similar to epinephrine [41].

BDNF and NGF are naturally secreted by urothelial and detrusor smooth muscle and are promising potential biomarkers for OAB. In patients with OAB, elevated urinary concentrations of NGF and BDNF have been shown to be valuable indicators for assessing the therapeutic efficacy of drugs used in pharmacotherapy. NGF and BDNF play an important role in the regulation of sensory neurons in OAB, contributing to normal neuronal function.

Current treatment options for OAB include antimuscarinic agents (such as oxybutynin, propiverine, solifenacin, tolterodine, trospium, darifenacin, and fesoterodine) as well as botulinum toxin and asiatic acid (as previously described) [2]. Similarly, in our present study, imperatorin demonstrated the ability to reduce urinary levels of BDNF and NGF in OAB patients (as shown in Figure 3). These findings highlight the potential of imperatorin as a promising candidate for the treatment of OAB and further support the need for further research in this area.

Recent research into the pathophysiology of overactive bladder (OAB) syndrome has discovered possible connections with urothelial barrier disruption. This disruption is often associated with ageing, inflammation, or infection and may impact detrusor permeability, leading to the presence of toxic urine components (such as high levels of urea and potassium) in combination with inflammatory mediators.

Abad et al. examined the effects of imperatorin on macrophage functions linked to inflammation. They suggested a pathway linking the release of enzymes (COX-1 and 5-LOX) and their resultant products (PGE2 and LTC4). Moreover, *E. coli* lipopolysaccharides triggered COX-2 and NOS activity [42].

Imperatorin demonstrated significant to moderate inhibition of PGE2 release within the 25–100 µmol dosage range and considerably reduced LTC4 formation within the 1–100 µmol range. Additionally, imperatorin had a noteworthy impact on 5-LOX release, with an IC50 value of under 15 µmol. Imperatorin doses ranging from 25–100 µmol effectively suppressed LPS-induced PGE2 release, which was calculated in ng/mL. The inhibition was associated with COX-2 activity and was dependent on the dosage administered.

These findings suggest that imperatorin may have a role in regulating inflammatory processes relevant to overactive bladder (OAB). This highlights the need for further research into its potential therapeutic benefits.

In the rats exposed to retinyl acetate, we found an increased concentration of vital bladder afferent neurotransmitters—CGRP (Figure 3)—in the urothelial tissue. These neuropeptides co-regulate smooth muscle contractions and are secreted in response to noxious stimuli, with a notable involvement in the inflammatory process [43]. CGRP levels are elevated in patients with OAB syndrome. It was found that after retinyl acetate instillation in rats, there was an increase in the levels of VAChT and OCT3 proteins involved in acetylcholine transport (Figure 3), like what was observed by Lips et al. (2007) [44]. As previously demonstrated, imperatorin has the potential to alleviate these extra-molecular pathological pathways. The reduced levels of VAChT in the detrusor muscle after IMP treatment (as shown in Figure 3) inhibit detrusor muscle contraction, leading to improved cystometric outcomes as described in the respective sections.

Our findings align with previous preclinical experiments and clinical trial reports. Our previous study showed that female Wistar-Kyoto rats with spontaneous hypertension and the RA-induced model of detrusor overactivity (DO)—diagnosed in 60–90% of patients with OAB [45]—exhibited similar changes in CGRP, OCT3, and VAChT levels, as reported in the study by Wróbel et al. (2020) [2]. Likewise, Smet et al. (1997) [43] observed a heightened occurrence of nerve fibers with CGRP in the urinary bladder of individuals with detrusor instability. In contrast, while our study revealed a noteworthy reduction in the urothelium barrier function of animals exposed to retinyl acetate, Liu et al. (2012) [36] did not identify such an abnormality in patients with OAB.

The control of the bladder is a complex process that involves multiple levels. Our study aimed to examine the impact of imperatorin on c-Fos expression in the neuronal voiding centers, namely the MPA, PMC, and vlPAG (as shown in Figure 4). c-Fos expression, a marker of neuronal activity [46], is central to understanding the role of PMC and PAG in the supraspinal control of continence and micturition [47]. Bladder stimuli in OAB activate the central micturition regions, resulting in an increase in c-Fos expression, as described by Kim et al. (2012) [48]. In addition, their research discovered significantly raised c-Fos expression in the neuronal voiding centers of the OAB animal model [48]. In our current study, we observed that RA induced significant c-Fos expression in all analyzed centers; however, the effect was mitigated by IMP (Figure 4). The decreased levels of c-Fos could signify symptom relief caused by OAB and may partially elucidate one of the potential mechanisms of imperatorin’s action. In addition, we investigated the expression of the multipotent enzyme ATP, which is implicated in various stages of molecular metabolism. This expression rose because of RA action and subsequently declined after IMP administration. Therefore, it is conceivable that imperatorin’s potent effect on restoring its levels in our experiments may be attributed to a multifactorial mechanism.

In conclusion, this study investigated imperatorin’s potential as a pharmacological intervention for overactive bladder syndrome and presented encouraging evidence supporting its efficacy as a treatment. The findings illustrate that imperatorin effectively ameliorated the pathological changes in bladder behavior associated with detrusor overactivity when administered intraperitoneally at a dose of 10 mg/kg/day. Imperatorin improved cystometric parameters, including voided volume, threshold pressure, and bladder capacity, while reducing the frequency and amplitude of nonvoiding contractions and bladder pressure. These effects suggest that imperatorin positively influences detrusor muscle tone and excitability during the storage phase of the micturition cycle. Importantly, imperatorin did not adversely affect urinary function, heart rate, or blood pressure, indicating its safety in this context. Biochemical analyses revealed that imperatorin reduced the levels of key biomarkers associated with overactive bladder, which suggests that imperatorin may modulate inflammatory processes, urothelial barrier function, and neural signaling pathways relevant to overactive bladder. Additionally, imperatorin reduced the expression of c-Fos in central micturition areas, indicating a potential mechanism for its action in relieving OAB symptoms.

One strong aspect of this study is its ability to offer valuable insights into the potential therapeutic benefits of imperatorin in a rat model of overactive bladder. This condition has a significant impact on quality of life, particularly in the elderly population. The study took a comprehensive approach, utilizing conscious cystometry, biochemical analyses, and the assessment of cardiovascular parameters. This allowed for a thorough evaluation of imperatorin’s impact on different aspects of bladder function. Additionally, imperatorin’s safety profile, evidenced by its absence of adverse effects on urinary function, heart rate, and blood pressure, increases its potential as a treatment option for patients with overactive bladder. However, the study has limitations, as it exclusively focuses on female rats, thereby restricting the generalizability of the findings to both genders. To address potential gender-specific effects, future research must include male rats. Although the study offers insights into imperatorin’s mechanism in the bladder, additional research is necessary to comprehend the specific pathways and receptors involved. Furthermore, future investigations should focus on determining the long-term safety of imperatorin and its efficacy in human clinical trials.

The authors have attempted to respond to as many inquiries as feasible. However, certain questions remain unresolved, and there is enormous scope for future investigation. Specifically, human-subject clinical trials are critical for validating the safety and effectiveness of imperatorin as a therapy for overactive bladder. It is hoped that additional research will examine gender variances in the reaction to imperatorin therapy, along with the ideal amount and duration of the medication. Additionally, insights into imperatorin’s mode of action and potential drug targets for the bladder can be obtained by examining the specific molecular and cellular mechanisms through which it exerts its effects. Furthermore, the durability of imperatorin’s therapeutic effects and its potential for combination therapy with existing treatments for overactive bladder should be assessed through long-term studies.

This study provides compelling evidence that imperatorin could be a valuable addition to treatment options for overactive bladder, with the potential to enhance bladder function and decrease symptoms. However, further research is crucial, including prioritizing human clinical trials and investigation into gender disparities and long-term safety, to substantiate and expand on these findings.

## 4. Materials and Methods

All applied procedures were performed in accordance with binding European law related to experimental studies on animal models and were approved by the Local Ethics Committee (number LKE323/22).

### 4.1. Animals

A total of 60 female Wistar rats, initially weighing 250 g, were utilized in the experiments. The rats were individually placed in metabolic cages (3700M071, Tecniplast, West Chester, PA, USA) located in environmentally controlled rooms with a temperature range of 22–23 °C, a natural light/dark cycle, and a relative humidity of approximately 45–55%. They were provided with unrestricted access to water and food throughout the duration of the study. The animals underwent a seven-day adjustment period under standardized conditions. The rats were randomly divided into four experimental groups, each consisting of 15 animals, as follows: (1) the control group received saline (CON); (2) the second group was administered retinyl acetate (RA); (3) the third group received imperatorin (IMP); (4) the fourth group was given a combination of retinyl acetate and imperatorin (RA + IMP).

### 4.2. Study Design

After completion of the surgical procedures, administration of IMP or vehicle continued for 14 days. Two days after the last dose of IMP, cystometric examination and assessment of bladder blood flow (BBF) were performed. The rats were then individually placed in metabolic cages (3700M071, Tecniplast) for 24 h. to assess urine production (UP), heart rate (HR), and mean arterial pressure (MAP). The animals were then humanely euthanized by decapitation, and biochemical analyses were performed.

### 4.3. Drugs Used in the Study

Retinyl acetate (Sigma-Aldrich) was diluted to a 0.75% solution using a mixture of Polysorbate 80 and saline. It was administered through intravesical instillation to induce bladder detrusor overactivity.

Imperatorin 9-(3-Methylbut-2-enyloxy)-7H-furo[3,2-g]chromen-7-one was obtained from Sigma-Aldrich (Cat. No. I6659; St. Louis, MO, USA). Imperatorin was suspended in a 1% solution of Tween 80 (Sigma, St. Louis, MO, USA) and dissolved in saline (0.9% NaCl). It was injected intraperitoneally (i.p.) at a volume of 10 mL/kg of body weight in a daily dose of 10 mg/kg for 14 consecutive days. The dosages of the administered compounds were determined based on the findings of previous studies conducted by our research team and were further validated and adjusted in preliminary experiments performed in our laboratory [35,49,50,51]. The control group received an equivalent volume of the vehicle as a matched dose.

### 4.4. Surgical Procedures

All surgical procedures were performed according to the previously described protocol [35,52]. Rats were anesthetized by intraperitoneal injection of 75 mg/kg ketamine hydrochloride (Ketanest; Pfizer Inc., New York, NY, USA) and 15 mg/kg xylazine (Sedazin; Biowet, Poland). Rats were placed in the supine position on a warming mattress set at 37 °C. Adequacy of anesthesia was determined by the absence of spontaneous movement and withdrawal response to a noxious toe pinch. After catheterization, a polyethylene catheter was inserted into the bladder through the external urethral orifice. Residual urine was removed, and either 0.75% retinyl acetate solution (to induce detrusor overactivity) or vehicle was instilled intravesically for five minutes. The bladder was then emptied, gently flushed with saline, and the catheter was removed. The abdominal wall was shaved and cleaned, and a vertical incision of approximately 10 mm was made. After careful dissection of the bladder from the surrounding tissue, a double-lumen polyethylene catheter (inner diameter, i.d., 0.28 mm; outer diameter, o.d., 0.61 mm; BD, Franklin Lakes, NJ, USA) filled with physiological saline and with a cuff at the end was inserted into the apex of the bladder dome. The catheter was secured with a 6-0 Vicryl suture. In addition, the carotid artery was cannulated with a polyethylene catheter (i.d. 0.28 mm; o.d. 0.61 mm; BD) filled with 40 IU/mL heparinized physiological saline for blood pressure measurement. The catheters were tunneled subcutaneously and exited in the retroscapular area, where they were connected to a plastic adapter to prevent removal by the animal. Healon (Pharmacia A.B., Uppsala, Sweden) was applied around the bladder at a dose of 0.85 mL to prevent adhesion. The abdominal incision was closed in layers, with the anatomical layers sutured using 4/0 catgut sutures. Silk ligatures were used to seal the free ends of the catheters. Animals received a subcutaneous injection of 100 mg cefazolin sodium hydrate (Biofazolin; Sandoz, Poland) to prevent urinary tract infection.

### 4.5. Conscious Cystometry

Cystometric examinations were performed 16 days after the surgical procedures, specifically 2 days after the last administration of IMP, as previously described [3,53]. The bladder catheter was connected to a pressure transducer (FT03; Grass Instruments, West Warwick, RI, USA) positioned at the bladder level and to a microinjection pump (CMA 100; Microject, Solna, Sweden) to monitor intravesical pressure and to infuse physiological saline into the bladder. Conscious cystometry was performed by gradually filling the bladder with physiological saline at a constant rate of 0.05 mL/min (equivalent to 3 mL/h) at room temperature (22 °C) to induce repetitive voiding. The infusion rate was determined based on preliminary studies in which rates between 0.05 and 0.1 mL/min produced cystometry profiles similar to those observed in the intact lower urinary tract of rats. The analogue signal from the pressure transducer was amplified and digitized using the Polyview system (Grass Instruments). Micturition volumes were measured using a fluid collector connected to a force-displacement transducer (FT03C; Grass Instruments). Both transducers were connected to a polygraph (7 DAG; Grass Instruments). Cystometry profiles and micturition volumes were continuously recorded on a Grass polygraph (Model 7E; Grass Instruments) and analyzed graphically. Data were sampled at a rate of 10 samples per second. Measurements for each animal represented the average of five micturition cycles obtained during repetitive voiding. Mean values from all animals in each condition were pooled to generate collective data for each condition. All procedures were performed by a person blinded to the treatments.

### 4.6. Bladder Blood Flow (BBF)

Immediately after bladder emptying, blood bladder flow (BBF) was assessed five times for each rat bladder using a laser Doppler blood perfusion imager (PeriScan PIM III, Perimed AB, Stockholm, Sweden). The BBF measurements were displayed as variations in laser Doppler frequency, represented by a color scale.

### 4.7. The Assessment of Cardiovascular Parameters and Diuresis

Following completion of cystometry, the tested animals were individually placed in metabolic cages for a period of 24 h. to assess the effect of IMP on mean arterial pressure (MAP), heart rate (HR), and daily urine production (UP).

### 4.8. Determining the Expression Levels of c-Fos in Central Micturition Areas

Using the stereotactic atlas of the rat’s brain and with the bregma serving as the reference point, the periaqueductal gray matter surrounding the central canal (PMC defined as the region spanning from bregma −9.68 to −9.80 mm), ventrolateral periaqueductal gray (vlPAG encompassed the region from bregma −7.64 to −8.00 mm), and medial preoptic area (MPA spanned from bregma −0.26 to 0.80 mm) were carefully isolated. On average, ten sections per region were obtained from each rat [54].

### 4.9. Biochemical Analyses

Levels of the following biomarkers were determined in the bladder urothelium: calcitonin gene-related peptide (CGRP; Biomatik, CN EKU02858, Kitchener, Ontario, Canada); organic cation transporter 3 (OCT3; antibodies-online, CN ABIN6227163, Limerick PA, USA); transient receptor potential cation channel subfamily V, member 1 (TRPV1; LSBio, LS-F36019, Poznan, Poland); ATP citrate lyase (ATP; LifeSpan BioSciences, LS-F10730, Seattle, WA, USA); malondialdehyde (Biomatik, CN EKF57996, Seattle, WA, USA); and 3-nitrotyrosine (LifeSpan BioSciences; CN LS-F40120-1, Seattle, WA, USA). The level of vesicular acetylcholine transporter (VAChT; LifeSpan BioSciences, CN LS-F12924-1, Seattle, WA, USA) and Rho kinase (ROCK1; LifeSpan BioSciences, LS-F32208, Seattle, WA, USA) has been marked in bladder detrusor muscle while the concentration of nerve growth factor (NGF; LifeSpan BioSciences, CN LS-F25946-1, Seattle, WA, USA) and brain-derived neurotrophic factor (BDNF; PROMEGA, CN G7610, Walldorf, Germany) were measured in the urine. c-Fos (c-Fos; MyBioSource, MBS729725, San Diego, CA, USA) expression was measured in the central micturition areas—medial preoptic area (MPA), ventrolateral periaqueductal gray (vlPAG), and pontine micturition center (PMC). All measurements were carried out according to the manufacturers’ instructions. Each sample was measured in duplicate. The results are presented in pg/mL.

### 4.10. Statistical Analysis

The obtained data were analyzed by the one-way ANOVA followed by Tukey’s multiple comparisons test. The normal distribution of the data was examined with the W Shapiro–Wilk test, and the equality of variance was tested with the Brown–Forsythe test. All results are presented as the mean ± standard error of the mean (SEM). A two-sided significance level (*p* value) of less than 0.05 was considered statistically significant with 95% confidence. All statistical analyses were performed using GraphPad Prism version 10.0.0 for Windows, GraphPad Software, Boston, MA, USA.

## Figures and Tables

**Figure 1 ijms-24-15793-f001:**
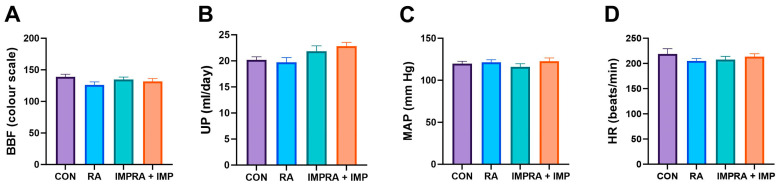
Effect of imperatorin on bladder blood flow and cardiovascular and diuresis parameters in the rapamycin-induced overactive bladder rat model. (**A**) bladder blood flow; (**B**) urine production; (**C**) arterial pressure; (**D**) heart rate. Data presented as means and SEM.

**Figure 2 ijms-24-15793-f002:**
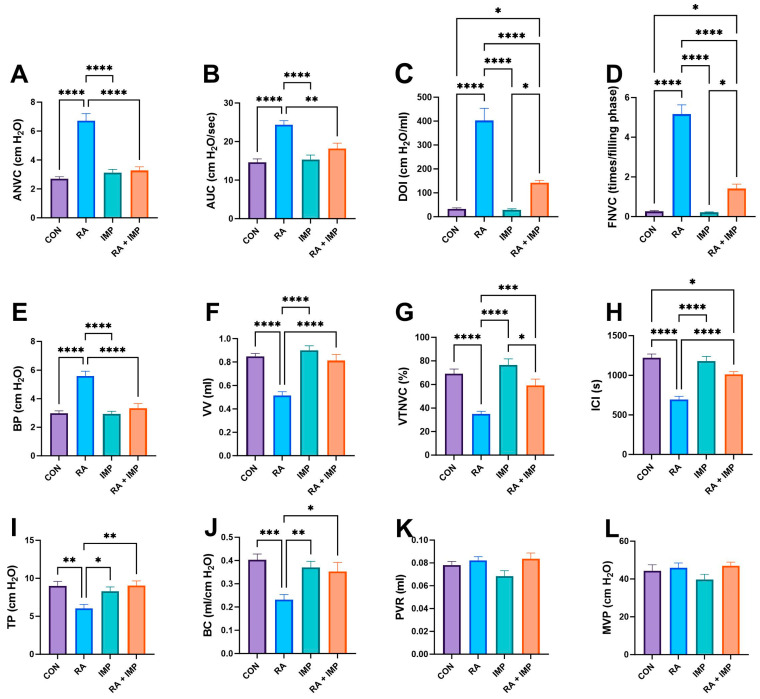
Effect of imperatorin on conscious cystometry parameters in the retinyl acetate-induced overactive bladder rat model. (**A**) nonvoiding contraction amplitude; (**B**) area under the pressure curve; (**C**) detrusor overactivity index; (**D**) nonvoiding contraction frequency; (**E**) basal pressure; (**F**) voided volume; (**G**) volume threshold to elicit nonvoiding contractions (NVC); (**H**) intercontraction interval; (**I**) threshold pressure; (**J**) bladder compliance; (**K**) post-void residual; (**L**) micturition voiding pressure. Data presented as means ± SEM. * *p* < 0.05, ** *p* < 0.01, *** *p* < 0.001, **** *p* < 0.0001.

**Figure 3 ijms-24-15793-f003:**
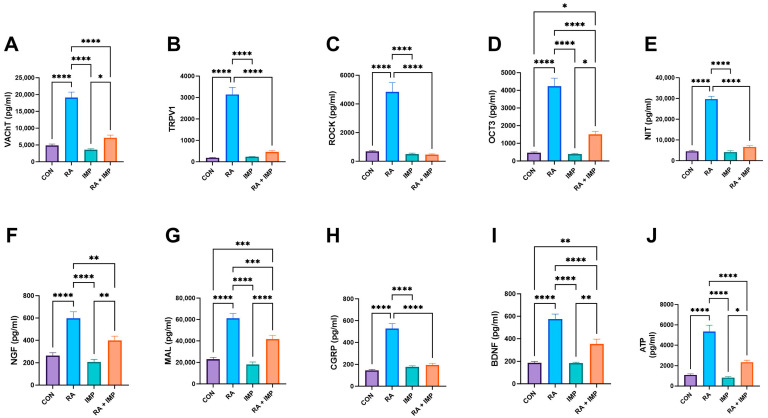
Effect of imperatorin on selected biochemical parameters in the retinyl acetate-induced overactive bladder rat model. (**A**) vesicular acetylcholine transporter; (**B**) transient receptor potential cation channel subfamily V, member 1; (**C**) Rho kinase; (**D**) organic cation transporter 3; (**E**) 3-nitrotyrosine; (**F**) nerve growth factor; (**G**) malondialdehyde; (**H**) calcitonin gene-related peptide; (**I**) brain-derived neurotrophic factor; (**J**) adenosine triphosphate. Data presented as means ± SEM. * *p* < 0.05, ** *p* < 0.01, *** *p* < 0.001, **** *p* < 0.0001.

**Figure 4 ijms-24-15793-f004:**
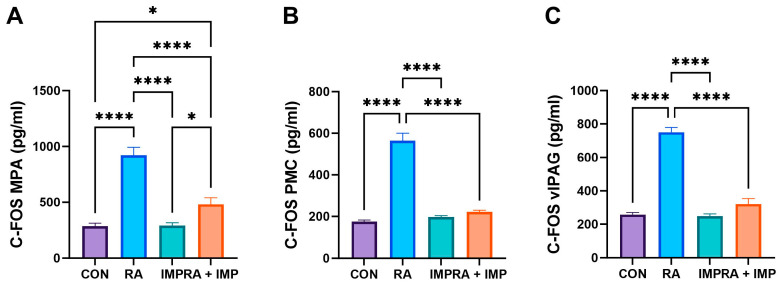
Effect of imperatorin on c-Fos in the retinyl acetate-induced overactive bladder rat model. (**A**) AP-1 transcription factor subunit in medial preoptic area; (**B**) AP-1 transcription factor subunit in periaqueductal gray matter; (**C**) AP-1 transcription factor subunit ventrolateral in ventrolateral periaqueductal gray matter. Data presented as means ± SEM. * *p* < 0.05, **** *p* < 0.0001.

## Data Availability

The data presented in this study are available on request from the corresponding author.

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
