# Peer review of "The Assessment of the Efficacy of Imperatorin in Reducing Overactive Bladder Symptoms"

_ijms, 2023, doi:10.3390/ijms242115793_

Round 1

Reviewer 1 Report

Comments and Suggestions for Authors

The paper investigates the effects of imperatorin, a natural compound, on bladder function and overactive bladder (OAB) symptoms in rats. The authors used intravesical infusion of retinyl acetate (RA) to induce detrusor overactivity, a common feature of OAB, in female Wistar rats. They administered imperatorin intraperitoneally at a dose of 10 mg/kg/day for 14 days and performed conscious cystometry to measure bladder pressure, voiding volume, and other parameters. The paper finds that imperatorin effectively normalizes the pathological changes in bladder function induced by RA, such as increased non-voiding contractions, decreased bladder capacity, and reduced voiding efficiency. The paper also finds that imperatorin reduces the levels of various biomarkers related to bladder afferent neurotransmission, inflammation, and nerve growth in the urothelium and detrusor muscle. Moreover, imperatorin decreases the expression of c-Fos, a neuronal activity marker, in the central micturition areas of the brain. Specific comments:

1.          The paper is well-written and organized, with clear objectives, methods, results, and discussion sections. The paper provides sufficient background information and literature review on OAB and imperatorin. The paper also uses appropriate statistical analysis and citation style.

2.          The introduction section should provide more rationale and justification for the choice of imperatorin as a potential treatment for OAB. What are the advantages and disadvantages of imperatorin compared to other natural or synthetic compounds? How does imperatorin interact with other drugs or herbal medicines that are used for OAB?

3.          The methods section should describe the source and purity of imperatorin and RA, as well as the dose and route of administration. How were these parameters determined?

4.          The discussion section should interpret the results in light of the existing literature and explain the possible mechanisms of action of imperatorin in the bladder. How does imperatorin affect the urothelium, detrusor muscle, nerve growth factors, neurotransmitters, inflammatory mediators, and central micturition areas? What are the similarities and differences between imperatorin and other pharmacological agents for OAB? What are the implications and limitations of the study for clinical practice and future research?

5.          Are there any potential side effects of herb-based approach? Please discuss the safety of the herbal preparation.

6.          The conclusion section should summarize the main findings and contributions of the study, as well as suggest directions for further investigation. What are the strengths and weaknesses of the study? What are the unanswered questions or challenges that remain to be addressed?

Author Response

Answers to reviewer comments

Reviewer 1

The paper investigates the effects of imperatorin, a natural compound, on bladder function and overactive bladder (OAB) symptoms in rats. The authors used intravesical infusion of retinyl acetate (RA) to induce detrusor overactivity, a common feature of OAB, in female Wistar rats. They administered imperatorin intraperitoneally at a dose of 10 mg/kg/day for 14 days and performed conscious cystometry to measure bladder pressure, voiding volume, and other parameters. The paper finds that imperatorin effectively normalizes the pathological changes in bladder function induced by RA, such as increased non-voiding contractions, decreased bladder capacity, and reduced voiding efficiency. The paper also finds that imperatorin reduces the levels of various biomarkers related to bladder afferent neurotransmission, inflammation, and nerve growth in the urothelium and detrusor muscle. Moreover, imperatorin decreases the expression of c-Fos, a neuronal activity marker, in the central micturition areas of the brain. Specific comments:

The paper is well-written and organized, with clear objectives, methods, results, and discussion sections. The paper provides sufficient background information and literature review on OAB and imperatorin. The paper also uses appropriate statistical analysis and citation style.

1.The introduction section should provide more rationale and justification for the choice of imperatorin as a potential treatment for OAB. What are the advantages and disadvantages of imperatorin compared to other natural or synthetic compounds? How does imperatorin interact with other drugs or herbal medicines that are used for OAB?

Imperatorin, a furanocoumarin derivative, has many documented pharmacological properties that make it a candidate for potential drug development. Several important biological properties have been described which suggest that imperatorin is an important bioactive molecule and can be considered as a possible structure for further drug modelling and development. Imperatorin may be useful in the treatment of many disorders such as epilepsy, anxiety, depression, and as an AChE inhibitor it has prospects in the treatment of Parkinson's and Alzheimer's diseases. Other important biological properties have also been attributed to imperatorin, including antibacterial, anticancer, anti-osteoporotic, myorelaxant, anti-inflammatory, cardioprotective, hepatoprotective and inhibitor of HIV replication. It may also merit further biological and pharmacological evaluation for the treatment of cardiovascular disease, osteoporosis, cancer, and overactive bladder. We have added additional references to support these characteristics and the possible effects of imperatorin.

In addition, a strong aspect of our study is its ability to provide valuable insights into the potential therapeutic benefits of imperatorin in a rat model of overactive bladder. This condition has a significant impact on quality of life, particularly in the elderly population. The study took a comprehensive approach using conscious cystometry, biochemical analysis and assessment of cardiovascular parameters. This allowed a thorough evaluation of the effect of Imperatorin on various aspects of bladder function. In addition, the safety profile of imperatorin, as evidenced by the absence of adverse effects on urinary function, heart rate and blood pressure, increases its potential as a treatment option for patients with overactive bladder. (Lines 343-351 of the manuscript)

  1. The methods section should describe the source and purity of imperatorin and RA, as well as the dose and route of administration. How were these parameters determined?

Two substances were used in the presented research: Retinyl acetate and Imperatorin. The "Drugs used in the study" section, which is part of the "materials and methods" section, contains information about the origin of the substance. Both were purchased from Sigma-Aldrich. Their purity is ≥ 98%. This section also contains information on the doses and route of administration of the above substances.

“Retinyl acetate (Sigma-Aldrich) was diluted to a 0.75% solution using a mixture of Polysorbate 80 and saline. It was administered through intravesical instillation to induce bladder detrusor overactivity. Imperatorin 9-(3-Methylbut-2-enyloxy)-7H-furo[3,2-g]chromen-7-one was obtained from Sigma-Aldrich (Cat. No. I6659; St. Louis, USA). Imperatorin was suspended in a 1% solution of Tween 80 (Sigma, St. Louis, MO, USA) and dissolved in saline (0.9% NaCl). It was injected intraperitoneally (i.p.) at a volume of 10 mL/kg of body weight in a daily dose of 10 mg/kg for 14 consecutive days. The dosages of the administered compounds were determined based on the findings of previous studies conducted by our research team and were further validated and adjusted in preliminary experiments performed in our laboratory [27,41–43]. The control group received an equivalent volume of the vehicle as a matched dose.”

3.The discussion section should interpret the results in light of the existing literature and explain the possible mechanisms of action of imperatorin in the bladder. How does imperatorin affect the urothelium, detrusor muscle, nerve growth factors, neurotransmitters, inflammatory mediators, and central micturition areas? What are the similarities and differences between imperatorin and other pharmacological agents for OAB? What are the implications and limitations of the study for clinical practice and future research?

The proposed mechanism of action of imperatorin is both central and peripheral, as clearly evidenced by the reduced of the main marker of the central micturition center, c-Fos.

The central mechanism of action of imperatorin is to inhibit the activity of centers responsible for micturition: the pontine micturition center (PMC, defined as the region spanning from bregma -9.68 to -9.80 mm); ventrolateral periaqueductal gray (vlPAG, encompassed the region from bregma -7.64 to -8.00 mm), and medial preoptic area (MPA, spanned from bregma -0.26 to 0.80 mm), as evidenced by a decrease in c-Fos shown in our study, which is a marker of the biogenic activity of the mentioned centers.

The peripheral mechanism of action of imperatorin appears to be multidirectional. The performed biochemical analysis showed a reduction in peripheral cholinergic transmission, as evidenced by a decrease in the activity of the organic cation transporter 3 (pg/ml) (OCT3, determined in the urinary bladder epithelium) and the vesicular acetylcholine transporter (pg/ml) (VACht, marked in the detrusor muscle of the bladder), which are involved in acetylcholinergic transmission. Other mechanisms include a reduction in rho-kinase activity (pg/ml) (ROCK) marked in the detrusor muscle of the bladder and expression of the multipotent enzyme ATP (determined in the bladder urothelium) which is implicated in various stages of molecular metabolism. This expression increased under the influence of retinyl acetate and then decreased after the administration of imperatorin. Therefore, it is conceivable that the strong effect of imperatorin on restoring its levels in our experiments may be attributable to a multifactorial mechanism. In our study, we also proved that imperatorin reduces the level of malondialdehyde (MAL, pg/ml) and 3-nitrotyrosine (NIT, pg/ml), markers of the oxidation process. Moreover, as we have shown, imperatorin was found to reduce the sensitivity of TRPV1 sensory receptors in the bladder epithelium. Furthermore, we have shown that imperatorin reduces the level of the markers brain-derived neurotrophic factor (BDNF, pg/ml) and nerve growth factor (NGF, pg/ml), promising potential biomarkers of OAB and naturally secreted by the urothelium and detrusor smooth muscle, which helps reduce OAB symptoms.

  1. 4. Are there any potential side effects of herb-based approach? Please discuss the safety of the herbal preparation.

Over the last thirty years, the consumption of herbal medicinal products and supplements has increased enormously, with at least 80% of people worldwide using them as a part of their primary care. Although therapies using these agents have shown promising potential and the effectiveness of many herbal products has been clearly established, many of them remain unproven and their use is either poorly monitored or even not monitored at all. Adverse events resulting from the consumption of herbal medicines can be attributed to several factors, which include mistaken use of the wrong plant species, adulteration of herbal products with other, undeclared drugs, contamination with toxic or hazardous substances, overdose, misuse of herbal medicines by healthcare providers or consumers and the use of herbal medicines with other medicines.

Ekor M. The growing use of herbal medicines: issues relating to adverse reactions and challenges in monitoring safety. Front Pharmacol. 2014 Jan 10; 4:177. doi: 10.3389/fphar.2013.00177. PMID: 24454289; PMCID: PMC3887317.

5.The conclusion section should summarize the main findings and contributions of the study, as well as suggest directions for further investigation. What are the strengths and weaknesses of the study? What are the unanswered questions or challenges that remain to be addressed?

Thank you for your valuable feedback. We appreciate your suggestions for improving the conclusion section. In response, we have revised the conclusion to summarize more comprehensively the main findings and contributions of the study. We have also addressed the strengths and weaknesses of our research and highlighted the unanswered questions and challenges that warrant further investigation. Your input has helped us to improve the overall quality and clarity of our manuscript. All the improvements and changes mentioned are made at the end of the discussion section.

Reviewer 2 Report

Comments and Suggestions for Authors

Dear authors,

Thank you for your creative research idea. However, there are some issues to be improved, before your work could be eligible for publication.

- there is no or very limited documentation about the preparation and selection of rats before the evaluation of their OAB symptoms. For example, we know that even in animal models, stress could be a crucial factor inducing OAB. So, you could describe the methodological steps before enrolling the rats.

- there is very limited reference about the systematic side effects IMP. They are needed to be extended in the Discussion section and in the Results as well, especially if you have your own data and specific endpoints on them.\

- from the past literature, we now that IMP could be beneficial even i neurological diseases. Please, define if you have any data about IMP passing the brain barrier, especially comparing to the classic anticholinergics. 

Round 2

Reviewer 2 Report

Comments and Suggestions for Authors

Accepted in the revised version.